# Interactions between Sodium Hyaluronate and β-Cyclodextrin as Seen by Transport Properties

**DOI:** 10.3390/ijms24032889

**Published:** 2023-02-02

**Authors:** Lenka Musilová, Aleš Mráček, Eduarda F. G. Azevedo, Artur J. M. Valente, Ana M. T. D. P. V. Cabral, Ana C. F. Ribeiro, Miguel A. Esteso

**Affiliations:** 1Department of Physics and Materials Engineering, Faculty of Technology, Thomas Bata University in Zlín, Vavrečkova 275, 760 01 Zlín, Czech Republic; 2Centre of Polymer Systems, Thomas Bata University in Zlín, tř. Tomáše Bati 5678, 760 01 Zlín, Czech Republic; 3Department of Chemistry, Centro de Química, University of Coimbra, 3004-535 Coimbra, Portugal; 4Faculdade de Farmácia, Universidade de Coimbra, 3000-548 Coimbra, Portugal; 5U.D. Química Física, Universidad de Alcalá, 28805 Alcalá de Henares (Madrid), Spain; 6Faculty of Health Sciences, Universidad Católica de Ávila, Calle Los Canteros s/n, 05005 Ávila, Spain

**Keywords:** sodium hyaluronate, β-cyclodextrin, diffusion, salting-in, transport properties, viscosity, diffusion coefficients

## Abstract

Knowledge of mass transport parameters, diffusion, and viscosity of hyaluronic acid (HA) in the presence of cyclodextrins is of considerable importance for areas such as food packaging and drug delivery, among others. Despite a number of studies investigating the functionalization of HA or the corresponding sodium salt by cyclodextrins, only a few studies have reported the effect of cyclodextrins on the mass transport of HA in the presence of these oligosaccharides. Here, we report the tracer binary and ternary interdiffusion coefficients of sodium hyaluronate (NaHy) in water and aqueous β-cyclodextrin solutions. The diffusion behavior of sodium hyaluronate was dependent on the reduced viscosity of NaHy, which, in turn, presented a concave dependence on concentration, with a minimum at approximately 2.5 g dm^−3^. The significant decrease in the limiting diffusion coefficient of NaHy (at most 45%) at NaHy concentrations below 1 g dm^−3^ in the presence of β-cyclodextrin, taking water as the reference, allowed us to conclude that NaHy strongly interacted with the cyclodextrin.

## 1. Introduction

Hyaluronic acid (HA), also called hyaluronan, is a linear polysaccharide consisting of β-1,4-D-glucuronic acid and β-1,3 *N*-acetyl-D-glucosamine, which occurs naturally in all vertebrates. In humans, hyaluronic acid occurs mostly in the skin (approximately 50%) [1] but it can also be found in the umbilical cord or synovial fluid, and its function is to lubricate and transport nutrients to, for example, mobile joint structures [2]. The high hydrophilicity and viscoelastic properties of HA, associated with its natural role in the human body, have enabled the development of HA-based cosmetic and dermatological formulations in the last decade [3,4]. These formulations are often based on the formation of hydrogels, which overcomes the limitation of hyaluronic acid having low mechanical and chemical stability [5].

Despite the advantages of the formation of HA hydrogels, their high hydrophilicity limits their application as a matrix for drug delivery due to their low solubility. However, due to the presence of reactive functional groups (e.g., carboxylic and hydroxyl groups), HA can be chemically modified to increase the solubility of active substances. Cyclodextrins are among the species that can be grafted onto HA [6]. The obtained materials, with improved properties, are applied in areas related to cosmetics [7,8], drug and gene delivery [9,10,11,12], and biomedicine [13,14,15].

Cyclodextrins are oligosaccharides with amphiphilic properties; i.e., they are water soluble but possess a hydrophobic cavity as a result of an ether-bonding ring. This property allows the formation of host-guest supramolecular compounds with hydrophobic or amphiphilic molecules [16]. This remarkable property enables the use of cyclodextrins in fields such as the pharmaceutical industry [17,18,19], biomedicine [20], food packaging [21], water remediation [22,23] and sensors [24,25].

Despite the enormous interest that these compounds have generated in the last decades, and considering that important insights have been obtained by manipulating the rates of diffusion of carrier-drug complexes, it is intriguing that only a few studies have focused on the transport properties of hyaluronic acid or its salt, sodium hyaluronate [26,27,28]. In the present work, we measured the tracer and diffusion coefficients of sodium hyaluronate in water and aqueous β-cyclodextrin solutions, using the Taylor dispersion technique, to assess how the presence of cyclodextrin may affect the flux of NaHy [29]. These studies were complemented by measuring the viscosity of multicomponent systems.

## 2. Results and Discussion

### 2.1. Analysis of Viscosity Data

Viscosimetry is an efficient method suitable for assessing the effect of cyclodextrin on the structure of aqueous solutions of sodium hyaluronate. The viscosity of diluted polymer solutions can be expressed by the limiting viscosity number (LVN), [*η*], commonly calculated according to the Huggins equation through extrapolation of reduced viscosity *η_red_* versus concentration (*C*) dependence to zero concentration [30]. However, interesting information can be obtained about the behavior of polymers from the course of the viscosity curves, mainly in the case when the studied polymer belongs to the polyelectrolyte group, as in the case of NaHy. Therefore, in this work, viscosity measurements of NaHy in the concentration range of 0.1–10.0 g dm^−3^, in the absence and presence of β-cyclodextrin, were performed and the corresponding data are shown in Figure 1.

The dependence of the reduced viscosity, *η*_red_, on the two different NaHy concentration ranges (0.1–1 and 1–10 g dm^−3^, Figure 1) in the presence of βCD was assessed. It can be seen that, in general, the reduced viscosity decreased as βCD increased. Additionally the dependence of the reduced viscosity as a function of NaHy was not monothonic; i.e., a more accentuated decrease in the reduced viscosity of NaHy-βCD occurred from 0.1 to 1 g dm^−3^. This suggested that βCD had a more pronounced effect on the unpacking of hyaluronan polymer chains at very low concentrations, and thus, the reduced viscosity of the solution increased. Consequently, the entities of NaHy offerred more frictional resistance to motion through the liquid, and the diffusion coefficient of these aqueous systems became smaller (see Figure 2 below). On the contrary, the reduced viscosity of NaHy-βCD increased in the NaHy concentration range of 1–10 g dm^−3^ (Figure 1). This may be justified by the type of interactions occurring between hyaluronan and βCD entities. βCD behaved as an "ion" at lower concentrations of hyaluronan; i.e. it seems that βCD exerted a screening effect on NaHy interactions. On the other hand, more van der Waals interactions were formed between NaHy and βCD chains at higher concentrations of hyaluronan, thus the reduced viscosity of this polyelectrolyte increased slightly.

By increasing the NaHy concentration (*C* > 1 g dm^−3^), the interactions between βCD and NaHy gradually decreased and the effect of the reduced viscosity became apparent (Figure 1). The dependence of *η*_red_ as a function of *C* depicted in Figure 1 afforded additional interesting information about the studied samples and proved that NaHy behaved as a typical polyelectrolyte [31]. This type of behavior was commonly characterized by a nonlinear variation of the *η*_red_ vs. *C* plot with an upward curvature at the lowest NaHy concentration, recorded mainly for NaHy dissolved in MilliQ water (Figure 1).

The variations in viscosity with concentration of NaHy in water and in water/βCD were similar. That is, NaHy chains shrunk in the presence of water or in βCD/water and their hydrodynamic volume decreased, resulting in a drop in the LVN. The effect of the polymer began to predominate over the effect of βCD, verifying that for the solution with the highest NaHy concentration (10.0 g dm^−3^), the value of the reduced viscosity was similar to that found for the solution of NaHy at the same composition without βCD. In these circumstances, it was expected that the interactions between NaHy-βCD entities were negligible compared to those between NaHy-water and NaHy-NaHy.

At a sufficiently high ionic strength, the charges due to the carboxylate groups on the NaHy chain were completely screened. Nevertheless, in water, these repulsions increased the hydrodynamic volume of the macromolecule, thus the viscosity increased and diffusion decreased.

It could be said that there was a threshold concentration of NaHy at which the predominance of polymer-polymer interactions changed to polymer-solvent interactions.

The higher viscosity of NaHy in water arose from the different conformation of the polymer chains in the solvent. While NaHy chains were stretched in water, they shrunk in the presence of salts and their hydrodynamic volume decreased, resulting in a drop in the LVN. Better packing of NaHy in salt solutions was caused by shielding of the electrostatic repulsion between similar charges located along the polymer chain. At a sufficiently high ionic strength, the charges due to the carboxylate groups on the NaHy chain were completely screened. Nevertheless, this repulsion in water increased the hydrodynamic volume of the coil, thus only slightly inceasing the viscosity. It should be stressed that the reported reduced viscosities (Figure 1) were in good agreement with those reported previously for NaHy+H_2_O solutions [27,32,33,34].

### 2.2. Diffusion Coefficients Analysis

#### 2.2.1. Tracer Binary Diffusion Coefficients of Sodium Hyaluronate

Tracer diffusion coefficients of sodium hyaluronate (NaHy) in water and the apparent tracer diffusion coefficients of NaHy in aqueous solutions of β-cyclodextrin at different concentrations were measured at 25.00 °C (Figure 2). The *D* values reported in Table 1, Table 2 and Table 3 were computed from four to six replicate dispersion profiles.

Table 1 shows the deviations between the limiting binary diffusion coefficients of NaHy in aqueous solutions containing βCD and the diffusion coefficients of NaHy in water.

From the analysis of tracer binary diffusion coefficients (Figure 2), it can be concluded that the presence of βCD had a significant effect on the transport of NaHy by diffusion. This effect was more significant for lower NaHy concentrations, reaching a maximum deviation value (45%) at NaHy 1.0 g dm^−3^. 

However, these deviations decreased with increasing NaHy concentrations, reaching a minimum for *C* = 5 g dm^−3^. For higher concentrations, there was again a slight increase in *D*. As was indicated by the viscosity data, these deviations suggested a significant interaction, mainly evidenced for low NaHy concentrations. 

It is also worth noting that for the highest NaHy concentration, the deviation Δ*D*/*D* decreased with increasing βCD concentration, becoming negligible for [βCD] = 7 × 10^−3^ mol dm^−3^. These tracer values approached the values of *D*(NaHy) in water (without βCD). In this particular situation, the intermolecular interactions between NaHy-NaHy and NaHy-H_2_O were shown to be predominant compared to those existing between NaHy and βCD. Support for this behavior was provided by the viscosity studies of the same systems (Figure 1). At the lowest concentrations of NaHy (i.e. *C* < 1 g dm^−3^), the presence of βCD promoted a salting-in effect, contributing to the contraction of the NaHy chain and inducing a significant increase in its chain stiffness. This effect was more evident when the concentration decreased, leading to an increase in the reduced viscosity, and consequently, the diffusion coefficient of this aqueous system became smaller. It can be concluded that the effect of dissolved βCD predominated over the effect of the polymer, leading to negligible NaHy-NaHy interactions compared to the prevailing NaHy-βCD interactions. Thus, NaHy behaved as a structure-making solute (also known as a kosmotropic solute) [35]. Having the ability to organize the water structure, this solute was strongly hydrated in solution and its presence could favor the formation of hydrophobic aggregates.

However, for more concentrated NaHy solutions, the effect of the polymer began to predominate over the effect of βCD. This was observed for the solution with the highest NaHy concentration (10.0 g dm^−3^), in which the value of the reduced viscosity was similar to the value found for the solution of NaHy at the same composition without βCD. In these circumstances, it was expected that the interactions between NaHy-βCD were negligible compared to those between NaHy-water and NaHy-NaHy.

In summary, it can be seen that at NaHy concentrations below 1 g dm^−3^, the presence of βCD led to a decrease in *D*, which can be justified by a strong association. At NaHy concentrations above 1 g dm^−3^ (the so-called polymer saturation point [36]), the diffusion coefficients increased because of the increased free NaHy [37]. Such an increase in *D* occurred until an excess of NaHy was observed and, consequently, the tracer diffusion coefficient of NaHy wasre independent of βCD concentration. 

#### 2.2.2. Tracer Ternary Diffusion Coefficients of Sodium Hyaluronate

The tracer ternary diffusion coefficients for NaHy (*D*_11_, *D*_12_, *D*_21_, *D*_22_) in aqueous βCD (7 × 10^−3^ mol dm^−3^) solutions are summarized in Table 2. *D*_ik(i,k = 1,2)_ data represent average values obtained from at least four replicate dispersion profiles. The main diffusion coefficients *D*_11_ and *D*_22_ were reproducible within ± 0.015 ×10^−9^ m^2^ s^−1^, whilst the cross-diffusion coefficients, *D*_12_ and *D*_21_, showed reproducibility within ± 0.050× 10^−9^ m^2^ s^−1^.

From the analysis of the cross diffusion coefficients presented in Table 2, we can conclude the following: (a) the cross-coefficient *D*_12_ values were almost zero within the precision of the measurements; and (b) *D*_21_ values were large and negative, indicating substantial counter-current coupled flows of βCD. As expected, NaHy(1) concentration gradients could not drive coupled flows of β-CD(2) in solutions that did not contain NaHy. However, the interactions between NaHy and βCD led to significant counter-current coupled flows of β-CD. In fact, whereas the values of the *D*_21_/*D*_11_ ratio showed that a mole of diffusing NaHy counter-transported at most 1.5 mol of βCD, the *D*_12_/*D*_22_ values showed that 1 mole of diffusing βCD co-transported at most 0.1 mol of NaHy. 

Concerning the main diffusion coefficients, *D*_11_ and *D*_22_, they essentially coincided with the tracer ternary diffusion coefficients of NaHy dissolved in supporting βCD solutions (Figure 2) and the corresponding binary values [38] (that is, deviations ≤3.0%). Regarding the latter, it should be noted that the addition of NaHy produced small relative changes in *D*_22_. It can be hypothesized that this may be due to similar mobilities between βCD and possible βCD-NaHy aggregates, as supported by the viscosity measurements (Section 2.1).

## 3. Materials and Methods

The characteristics of reagents and solvents used in the experimental work are detailed in Table 3.

**Table 3 ijms-24-02889-t003:** Reagents used in the experiments.

Chemical Name	Source	CAS Number	Mass Fraction Purity
Sodium hyaluronate	Contipro Ltd. (Dolní Dobrouč, Czech Republic) (Mw = 243 kDa)	9067-32-7	>0.99 ^1^
β-cyclodextrin	Sigma-Aldrich (water mass fraction of 0.131)	7585-39-9	>0.99 ^1^
water	Millipore-Q water (18.2 MΩ·cm at 25.0 °C)	7732-18-5	

^1^ as declared by the supplier.

### 3.1. Viscosity Measurements

For the viscosity measurements, a set of NaHy solutions were prepared at concentrations of 0.1, 0.25, 0.5, 0.7, and 1.00 g dm^−3^ by dissolving the corresponding polymer powder in Milli-Q water with continuous stirring (24 h), followed by dissolving at 25.0 °C for 24 h. Another set of NaHy solutions, at the same concentrations and following the procedure described above, were prepared using aqueous β-cyclodextrin solutions at concentrations of 0.001, 0.005, and 0.007 mol dm^−3^. 

Viscosity measurements were performed using an SI Viscoclock automated viscometer (Schott Instruments, Karlsbad, Germany) equipped with an Ubbelohde type capillary viscometer at 25.0 ± 0.1 °C. The mean flow time of the NaHy solutions through the capillary was calculated from five repeatable measurements. The reduced *η_red_* viscosities of the NaHy solutions were calculated by using the equation:(1)$$\eta_{red}=\eta_{rel}/C_{NaHy}$$
where *η_rel_* is the relative viscosity and *C_NaHy_* is the mass concentration of the NaHy.

The dependences between the reduced viscosity and concentration according to Huggins were then plotted and compared.

### 3.2. Diffusion Measurements

Mutual diffusion coefficients for the binary {NaHy + H_2_O} and ternary {NaHy + H_2_O + βCD} solutions measured in this work were computed using Fick’s equations [39]: (2)J=−D∇C,
(3)$$J_{1}=-D_{11}\nabla C_{1}-D_{12}\nabla C_{2},$$
(4)$$J_{2}=-D_{22}\nabla C_{2}-D_{21}\nabla C_{1},$$
where *C* is the molar concentration of the solute; *D* is the binary diffusion coefficient; *D*_11_ and *D*_22_ are the main ternary mutual diffusion coefficients and represent the flux of each component (*J*_1_) and (*J*_2_) produced by its own concentration gradient; and *D*_12_ and *D*_21_ are the cross-diffusion coefficients representing the flux of NaHy caused by the βCD concentration gradient (∇*C*_2_) and the flux of βCD caused by the concentration gradient of NaHy (∇*C*_1_), respectively.

Diffusion coefficients in systems treated as binary (NaHy + H_2_O) and ternary (NaHy + H_2_O + βCD) were measured using the Taylor technique. Details of the technique are well described in the literature [39,40,41,42,43,44]. Basically, the sample solution (0.063 cm^3^) was injected into a laminar carrier solution of slightly different composition. The whole technique was kept at 25.00 (±0.01) °C using an air thermostat. The broadened distribution of the dispersed samples was monitored at the tube outlet by a differential refractometer (Waters model 2410). The refractometer output voltage, *V*(*t*), was measured at 5 s intervals using a digital voltmeter (Agilent 34401 A). The dispersion profiles for the binary {NaHy(1) + H_2_O(2)} solutions were analyzed by fitting the following equation (Equarion (5)):
*V*(*t*) = *V*_0_ + *V*_1_*t* + *V*_max_ (*t*_R_/*t*)^1/2^ exp[−12*D*(*t* − *t*_R_)^2^/*r*^2^*t*],(5)
where *r* is the internal radius of the dispersion tube; *t*_R_ is the mean sample retention time; *V*_max_ is the peak height; and finally, *V*_0_ and *V*_1_ are the baseline voltage and baseline slope, respectively [39,40,41,42,43,44]. The values of the tracer diffusion coefficient for NaHy (component 1), *D*_1_, in aqueous solutions of βCD at different concentrations can be estimated by assuming that coupled diffusion does not occur. Thus, the NaHy tracer Taylor peaks are treated as simple binary ones and described by a single “pseudo-binary” tracer diffusion coefficient. 

Extensions of the Taylor technique were also used to measure the tracer ternary mutual diffusion coefficient of NaHy in aqueous βCD carrier solutions at different concentrations (that is, 0.001, 0.005, and 0.007 mol dm^−3^). Three solutions containing 1.00, 5.00, and 10.0 g dm^−3^ were injected into each carrier aqueous solution. It should be noted that the flux and injected solutions had the same βCD concentrations. The respective dispersion profiles were analyzed by fitting the following equation (Equation (6)):(6)$$V\left(t\right)=V_{0}+V_{1}t+V_{\max}{\left(\frac{t_{R}}{t}\right)}^{0.5}\left[W_{1}exp\left(-\frac{12D_{1}{\left(t-t_{R}\right)}^{2}}{r^{2}t}\right)+\left(1-W_{1}\right)exp\left(-\frac{12D_{2}{\left(t-t_{R}\right)}^{2}}{r^{2}t}\right)\right],$$
where *W*_1_ is the normalized pre-exponential factor and *D*_1_ and *D*_2_ are the eigenvalues of the ternary diffusion coefficient matrix. The detailed analysis of the dispersion profiles is given in literature [45,46].

## 4. Conclusions

The reduced viscosity of sodium hyaluronate was significantly modified by the presence of βCD. In particular, a significant increase in the reduced viscosity was observed for NaHy at a concentration of 0.1 g dm^−3^, which was similar to the effect that occurs when the ionic strength increases in a polyelectrolyte solution. Since βCD is not a non-associated electrolyte, this finding is intriguing and understanding the mechanism that prevents (or decreases) interchain repulsion is interesting. The analysis of the ternary intermolecular diffusion coefficients suggested, however, that βCD might have a crosslinking effect between hyaluronate chains rather than a simple screening effect. This was indicated by the negative cross-diffusion coefficient, *D*_21_, which characterized counter diffusion of βCD induced by NaHy and the magnitude of the diffusion coefficient suggested strong NaHy-βCD interactions. By increasing the NaHy concentration, in particular for higher concentrations (>1 g dm^−3^), the effect of βCD became negligible and the viscosity and diffusion changed only slightly, thus highlighting the importance of Hy^−^-Hy^−^ and Hy^−^-H_2_O interactions.

In the systems evaluated in this paper, the interplay between diffusion and viscosity prevailed and hyaluronate-hyaluronate interactions could be tuning by adding βCD and, consequently, their transport properties could also be modified.

## Figures and Tables

**Figure 1 ijms-24-02889-f001:**
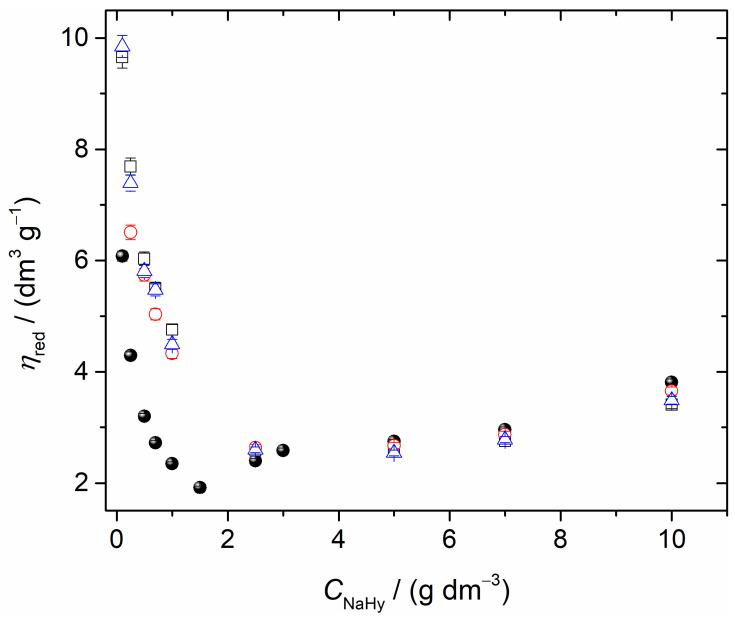
Dependence of reduced viscosity, *η*_red_, on the concentration of NaHy in water (●) and in solutions with different concentrations of β-cyclodextrin at 25 °C: (□) 0.001 mol dm^−3^; (o) 0.005 mol dm^−3^; and (Δ) 0.007 mol dm^−3^.

**Figure 2 ijms-24-02889-f002:**
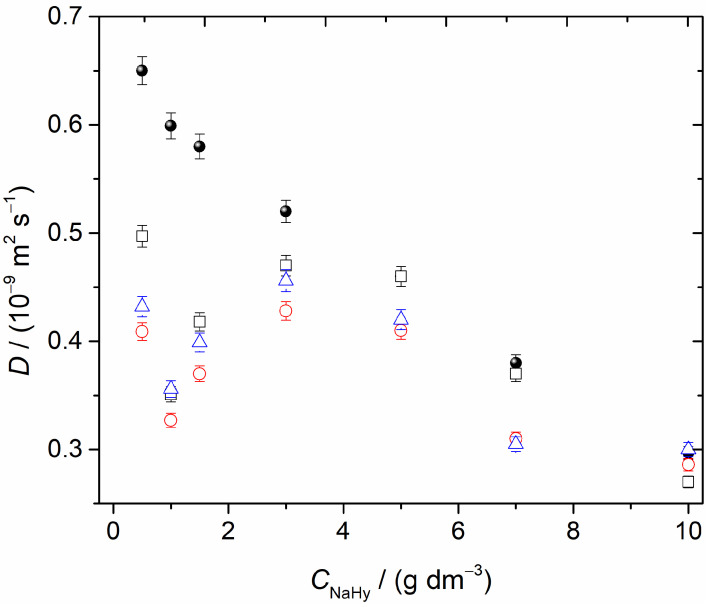
Tracer diffusion coefficients of NaHy in water (●) and in solutions with different concentrations of β-cyclodextrin at 25.00 °C: (□) 0.001 mol dm^−3^; (o) 0.005 mol dm^−3^; and (Δ) 0.007 mol dm^−3^.

**Table 1 ijms-24-02889-t001:** Relative deviations (Δ*D*/*D*%) between tracer diffusion coefficients of NaHy in aqueous solutions containing βCD and tracer diffusion coefficients of NaHy in water.

*C*_NaHy_/(g/dm^−3^)	Δ*D/D %*		
	[βCD] = 1 × 10^−3^ mol dm^−3^	[βCD] = 5 × 10^−3^ mol dm^−3^	[βCD] = 7 × 10^−3^ mol dm^−3^
0.50	−23.5	−37.1	−33.5
1.00	−41.0	−45.4	−40.6
1.50	−28.0	−36.2	−31.2
3.00	−9.6	−17.7	−12.3
5.00	0.0	−10.9	−8.7
7.00	−2.6	−18.4	−19.7
10.0	−9.1	−3.7	1.0

**Table 2 ijms-24-02889-t002:** Tracer diffusion coefficients (*D*_11_, *D*_12_, *D*_21_, *D*_22_) for NaHy (component 1) in aqueous βCD (component 2) solutions at C_2_ = 0.007 mol dm^−3^ and *T* = 25.00°C.

*C*_1_/(10^−3^ mol dm^−3^)	*C*_2_/(10^−3^ mol dm^−3^)	*D*_11_ ± *S*_D_/ (10^−9^ m^2^ s^−1^)	*D*_12_ ± *S*_D_/ (10^−9^ m^2^ s^−1^)	*D*_21_ ± *S*_D_/ (10^−9^ m^2^ s^−1^)	*D*_22_ ± *S*_D_/ (10^−9^ m^2^ s^−1^)
0.0 ^1^	7	0.430 ± 0.012	0.030 ± 0.010	−0.130 ± 0.030	0.380 ± 0.030
0.0 ^2^	7	0.476 ± 0.012	0.035 ± 0.010	−0.748 ± 0.010	0.368 ± 0.030
0.0 ^3^	7	0.443 ± 0.012	0.001 ± 0.010	−0.680 ± 0.010	0.389 ± 0.020

^1^ Injection of a solution of 1.0 g dm^−3^ concentration; ^2^ Injection of a solution of 5.0 g dm^−3^ concentration; ^3^ Injection of a solution of 10.0 g dm^−3^ concentration; Standard uncertainties are: *u*(*C*) = 5 ×10^−6^ (mol dm^−3^); *u*(*D*) ≅ 0.01×10^−9^ (m^2^ s^−1^); *u*(*T*) = 0.01 K. *D*_21_ are negative, indicating counter-current coupled flows.

## Data Availability

Data is contained within the article.

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
