# Peer review of "Interactions between Sodium Hyaluronate and β-Cyclodextrin as Seen by Transport Properties"

_ijms, 2023, doi:10.3390/ijms24032889_

Round 1
Reviewer 1 Report
Manuscript reports on a study on hyaluronan-cyclodextrin diffusional behavior. Methodologically it is based on established Taylor dispersion technique, the studied systems can be viewed as novel and of interest in medical applications. Experimental approach and data evaluation are thus standard and adequately applied. I thus have only a few notes before the final acceptance. Lines 258-259: the number of eq. should be (5) and also the remaining symbols of it should be explained. The concentration dependence of the viscosity of hyaluronan solutions was subject of various previous studies and readers thus should be informed on the comparison of results of this study and previous works.
Author Response
Manuscript reports on a study on hyaluronan-cyclodextrin diffusional behavior. Methodologically it is based on established Taylor dispersion technique, the studied systems can be viewed as novel and of interest in medical applications. Experimental approach and data evaluation are thus standard and adequately applied. I thus have only a few notes before the final acceptance.
Reply: We are grateful to the Reviewer#1 for her/his positive appreciation
Lines 258-259: the number of eq. should be (5) and also the remaining symbols of it should be explained.
Reply: The text was changed and now all symbols are explained (the majority of them were previously described after eq. (4)).
The concentration dependence of the viscosity of hyaluronan solutions was subject of various previous studies and readers thus should be informed on the comparison of results of this study and previous works.
Reply: The following sentence has been added: “It should be stressed that the reported reduced viscosities (Figure 2a) are in good agreement with those reported previously for NaHy+H2O solutions [27,32–34].”

Reviewer 2 Report
In this manuscript the authors are present the study of tracer and diffusion coefficients of sodium hyaluronate in water and in b-cyclodextrin aqueous solutions. They sought an answer to the question of how the presence of cyclodextrin can affect the flux of sodium hyaluronate. The results are show that the reduced viscosity of sodium hyaluronate is significantly modified by the precence of beta-CD.
I have found that the manuscript summarizes in good way the experimental results, conclusions drawn from data are good. The manuscript is well organised, I recommend it for publication after a minor revision.
- In the abstract, please explain what the abbreviation “HA” means.
- I recommend to change the order of the Section 2 and 3. “Materials and Methods” should be first.
Author Response
In this manuscript the authors are present the study of tracer and diffusion coefficients of sodium hyaluronate in water and in b-cyclodextrin aqueous solutions. They sought an answer to the question of how the presence of cyclodextrin can affect the flux of sodium hyaluronate. The results are show that the reduced viscosity of sodium hyaluronate is significantly modified by the precence of beta-CD.
I have found that the manuscript summarizes in good way the experimental results, conclusions drawn from data are good. The manuscript is well organised, I recommend it for publication after a minor revision.
Reply: We are grateful to the Reviewer#1 for her/his positive appreciation
- In the abstract, please explain what the abbreviation “HA” means.
Reply: We apologies for that. In the first line of the Abstract it is now written: “…hyaluronic acid (HA).”
- I recommend to change the order of the Section 2 and 3. “Materials and Methods” should be first.
Reply: We understand the Reviewer’s comment and I personally agree with that comment; however, we have followed the layout of the IJMS journal.
